# Gender Differences in Traumatic Experiences, PTSD, and Relevant Symptoms among the Iraqi Internally Displaced Persons

**DOI:** 10.3390/ijerph18189779

**Published:** 2021-09-16

**Authors:** Perjan Hashim Taha, Marit Sijbrandij

**Affiliations:** 1Psychiatry Unit, Department of Medicine, College of Medicine, University of Duhok, Duhok 42001, Kurdistan Region, Iraq; 2Department of Clinical, Neuro and Developmental Psychology, Amsterdam Public Health Research Institute, Vrije Universiteit, 1181 BT Amsterdam, The Netherlands; e.m.sijbrandij@vu.nl

**Keywords:** gender, trauma, PTSD, Iraq, internally displaced persons, refugees, humanitarian conflict

## Abstract

Conflict in Iraq has led to a large number of internally displaced Iraqis, with a great impact on their mental health. A few previous studies investigated the gender differences of mental disorders in Iraqi internally displaced persons (IDPs). The aim of this study was to assess gender differences among Iraqi IDPs after the 2014 terrorist attacks in terms of types of traumatic experiences, prevalence of post-traumatic stress disorder (PTSD), and other symptoms of common mental disorders (depression, anxiety, and somatization symptoms). A cross-sectional survey was conducted in April–June 2015 among 358 female and 464 male adult IDPs living in IDP camps in Duhok, Iraq. The Harvard Trauma Questionnaire (HTQ), General Health Questionnaire (GHQ-28), and Self-Reporting Questionnaire (SRQ-20) were applied by local interviewers. Comparison of scores of these measures between the two genders was performed using independent sample *t*-tests. Logistic regression analysis was carried out to identify predictors for PTSD. Although the types of traumatic events experienced by female and male IDPs were nearly similar, males reported higher exposure to combat situation, torture, oppressions, and destruction of personal properties (t = 3.718 and 4.758, respectively, *p* < 0.001). Overall, males experienced more events than females (*p* < 0.001). The probable PTSD prevalence rates (29.1% among females and 31.9% among males) did not differ significantly (*p* = 0.212). Female IDPs reported more somatic (*p* < 0.001) and depressive/anxious (*p* < 0.001) symptoms than males. The demographic factors and duration of camp stay were not associated significantly with PTSD diagnosis. Probable PTSD rates among male and female IDPs in Iraq are substantial. Although no gender differences were found in probable PTSD rates between female and male Iraqi IDPs, the mean score of common mental disorders cases was higher among females and they presented with higher levels of somatic and depressive/anxious symptoms. Further studies are needed to clarify the underlying mechanisms contributing to gender differences in PTSD.

## 1. Introduction

During the past decades, Iraq has gone through political instability with many internal and external conflicts. The 2014 events in Iraq—particularly the assaults of the terrorist organization called the Islamic State of Iraq and Syria (ISIS) [1] yielded a disastrous impact on the affected areas. The estimated number of internally displaced Iraqis during 2015 exceeded 3.19 million, spread across more than 3500 locations countrywide [2]. Displaced Iraqis are resettled in formal camps and in informal settlements across the country. The most tragic attacks were on the city of Sinjar and surrounding towns, which are mostly inhabited by Yazidi people [3]. They are a Kurdish religious minority living in the north of Iraq, west of Iran, east of Turkey, and north of Syria [4]. About 3100 Yazidis were killed and 6800 were kidnapped in these attacks [5]. During this period, the northern Iraq city of Duhok established seventeen camps for internally displaced persons (IDPs) that hosted about 200,000 persons [6].

In general, the wars and conflicts impact on the mental health of the affected population [7]. Subsequently, the incidence and prevalence of mental health disorders were increased in the IDP group [8]. The most common mental disorders (CMDs) among IDPs and refugees are post-traumatic stress disorder (PTSD), depression, and anxiety disorders [9]. A recent meta-analysis that examined the prevalence of mental disorders among global refugee populations showed that approximately 31% of refugees and asylum seekers displaced meet the criteria for PTSD, 32% for depression, and 11% for an anxiety disorder [10].

Generally, epidemiological studies across the world systematically reported higher rates of depression and anxiety disorders in women, whereas men demonstrate higher rates of substance use disorder [11]. PTSD prevalence is higher in females with a ratio of about 2:1 to males [12]. Additionally, research found that women are more strongly affected by war than men [7]. Female IDPs and refugees are highly vulnerable to certain mental disorders such as PTSD and depression [13]. Studies investigating the differences in PTSD incidence between male and female refugees are inconclusive. Some have shown that that the incidence of PTSD is significantly higher in women than in men, whereas other studies reported no differences between the presence of PTSD in men and women, but rather differences in terms of severity of PTSD symptoms [14]. Further, it has been reported that these differences in PTSD severity are more prominent after a number of years following resettlement [15]. Beyond PTSD symptoms, female asylum seekers described more emotional distress, somatic symptoms, and sexual dysfunction, while men reported higher levels of detachment [16]. In addition, the risk factors for these psychological symptoms have found to be different between both genders. Among males, unemployment and past traumatic experiences had the strongest contribution to PTSD, while among females and older age, Middle Eastern background, and living alone were important factors [17].

Explanations for the higher risk of PTSD in women than men include many theories. Biological theories posit that a genetic predisposition may be implicated in the increased risk for females as there is evidence of greater heritability of PTSD and depression among females [18,19]. Further, hormonal influences may play a role as testosterone, oestradiol, progesterone, and ALLO/5α-progesterone ratio are assumed to be concerned in the development of PTSD [18]. These factors work together to put females at a specific risk of developing PTSD either directly and/or through epigenetic mechanisms. On the other hand, the social construction of gender within the society may clarify the gender differences in PTSD, involving differences in trauma exposure and the risk for the development of PTSD symptoms [20]. Another explanatory framework is proposed by the relational cultural theory, recognizing sexism as a causative factor to young women’s increased risk for sexual violence [21]. Additionally, the type of traumatic experiences may differ between males and females, with females more often exposed to high-impact interpersonal, sexual, or gender-based trauma, and they are frequently subjected to these kinds of traumas early in their life, which is associated with a higher risk for PTSD [14,22]. Further, females display stronger perceptions of threat and perceive themselves as having less self-control in the acute phase following trauma than males [22,23]. In addition, it has been proposed that males and females have different cognitive schemas with female trauma victims more often blaming themselves for the trauma, holding more negative views of themselves and the world as being more dangerous than male trauma victims do [24]. It is yet unclear how females and males react differently to traumatic events and express different PTSD and other mental health symptoms and whether risk and protective factors differ between the two genders. Furthermore, the association of other symptoms of CMDs with exposure to traumatic events and PTSD in both genders has not been studied in internally displaced Yazidi refugees.

ISIS fighters were using different methods of violence toward males and females. Females who reached adolescence were often separated from males, after which they were used as slaves serving the ISIS commanders and were sexually abused repeatedly [25]. As a result, Iraqi IDP women and young girls present with high levels of PTSD [26]. A study of Yazidi women and children, 16% of whom had survived enslavement, found that over 80% met clinical criteria for a DSM-5 PTSD diagnosis [4]. ISIS fighters executed males older than 12 years old, kidnapped boys younger than 10–12, changed their religion and converted them to Islam, and put them under obligatory military training to become ISIS fighters [25]. In addition to exposure to these traumatic events by ISIS, IDPs are exposed to daily stressors in the displacement camps that further increase the risk for PTSD and other mental health problems [27]. These stressors include socioeconomic hardship, uncertainty about the future and about safety of family members. This is added to the fact that Iraqi IDPs have lost their houses, their jobs, and some of them their family members, relatives, and/or friends [28].

Only a few previous studies investigated gender differences of mental disorders among IDPs in Iraq. Our foci in this study were to address gender differences among Iraqi IDPs in the experienced traumatic events before, during, and after displacement, to examine PTSD and associated psychiatric symptoms, and to investigate the contributing factors to these gender differences.

## 2. Materials and Methods

### 2.1. Study Design and Setting

This study included a subsample of a dataset gathered among Iraqi IDPs and Syrian refugees investigating traumatic experiences [29]. The subsample of Iraqi IDPs was included in the current study. The cross-sectional study took place in April–June 2015 in the IDP camps in Duhok province. One camp, Khanke camp, was selected randomly from a total of 17 camps located 20 Km west of Duhok city. The camp opened on 6th of August 2014, and contained 3120 tents accommodating 16,460 individuals/2851 households [30]. Among the camp population, 53% were males and 47% were females. The study was approved by the scientific committee of the College of Medicine/University of Duhok and the Research Ethics Committee of Duhok Directorate of Health (reference number: 08042015-09-02).

### 2.2. Participants and Procedure

We included adult (≥18 years old) female and male IDPs from the population of camp inhabitants. IDPs having cognitive deficits, or disturbed insight or unable to communicate normally were excluded from the study. At first, the sample size was calculated according to this formula *n* = [z^2^·*p*·(1 − *p*)]/e^2^ where (z) is the critical standard score, (*p*) is population proportion, and (e) is margin of error. All tent numbers were entered into an excel sheet and 822 tents for IDPs were randomly selected for the interviewers to visit. Then, from the selected tents, one eligible adult from each family was invited to participate. Eight people refused to take part, and in that case another family member was selected randomly and asked to participate. The sample compromised 358 female and 464 male IDPs. Six counsellors were trained on the study tools by a psychiatrist with experienced in training of administration of such instruments. The face-to-face interviews endured for two months set out on 15 April 2015. During interviews, the interviewers promised the privacy of the gathered data.

### 2.3. Measures

Sociodemographic data that were assessed were gender, age, religion, marital status, educational level, employment, number of siblings, and duration of stay in the IDP camp.

The Harvard Trauma Questionnaire (HTQ) [31] measures the severity of trauma exposure, symptoms of PTSD [32]. We used part I–IV of the Iraqi version of the HTQ in this study [33]. Part I constitutes 43 items screening the experienced and witnessed traumatic events [34]. Part II screens for the most hurtful or terrifying experienced events. Part III assesses head injury. Part IV investigates the trauma-related symptoms; the first 16 items were deduced from the DSM-III symptoms of PTSD [35]. Items of Part IV are scores on a 4-point severity scale and the total score is divided by the number of items to get the total mean HTQ score. The HTQ part IV advocates a clinical cut-off score of 2.5 to indicate probable PTSD [36]. Several studies have demonstrated the validity and reliability of HTQ [32,35]. Cronbach’s alpha of HTQ Part IV was 0.90 in the present study, which suggested a good internal consistency.

The General Health Questionnaire (GHQ-28) is designed to screen for psychological well-being, emotional distress, and psychiatric morbidity [37]. It assesses somatic symptoms (items 1–7), anxiety/insomnia (items 8–14), social dysfunction (items 15–21), and severe depression (items 22–28) [38]. We used the validated Arabic language version [39]. A 4-point response score as well as a binary score can be computed (in which the first 2 responses are scored as 0 and the last 2 as 1) [40]. To screen for elevated psychological distress, by adding up the binary items, a cut-off score of 4 has been proposed [39,40]. In this study, the GHQ-28 showed a high internal consistency (Cronbach’s alpha = 0.82).

The Self-Reporting Questionnaire (SRQ-20) was used to examine CMDs. The SRQ-20 comprises of 20 yes/no questions concerning to 4 weeks prior the interview [41]. It consists of 4 subscales: depressive/anxious (4 items), somatic symptoms (6 items), reduced vital energy (6 items), and depressive thoughts (4 items) [42]. A cut-off score of 8 is widely used to determine common mental disorder cases [41,43]. In this study we applied a cut-off score of 7, which was used previously for Iraqi people [44]. It is a reliable instrument and has been validated previously in Arabic-speaking populations [45,46]. The internal consistency for testing reliability was good in this study (Cronbach’s alpha 0.82).

### 2.4. Statistical Analysis

The data were analyzed using the Statistical Package for Social Sciences (SPSS), version 22.0. The analysis of categorical data included Pearson’s chi-squared tests and percentages for the qualitative data. The comparison between HTQ, GHQ-28, and SRQ-20 scores between the two genders was performed using independent sample *t*-tests. Logistic regression analysis was adopted to determine the predictors for PTSD symptoms assessed with the HTQ among female and male Iraqi IDPs separately. PTSD measured by HTQ was the dependent variable, and the sociodemographic factors, duration of camp stay, and the number of experienced traumatic events were the independent variables. After a Bonferroni adjustment, it was decided that *p*-values smaller than 0.002 indicate statistical significance.

## 3. Results

### 3.1. Demographic Characteristics and PTSD Rates

The sociodemographic characteristics of the participants are demonstrated in Table 1. In our sample, 464 (56.4%) males constituted slightly larger proportion than participated females (*n* = 358; 43.6%). No significant differences were found between males and females regarding the following sociodemographic domains: age, religion, marital status, number of siblings, and duration of stay in camps. Three quarters of the respondents were young adults (18–40 years old) and married, and more than 98% were of the Yazidi religion. Illiteracy was found to be approximately twice as high among females (60.9%) comparing with males (31.5%; *p* < 0.001). The higher proportion of female IDPs (98%) were unemployed compared with 77.6% of males (*p* < 0.001). The probable PTSD prevalence rate assessed with the HTQ part IV was 29.1% among female IDPs and 31.9% among male IDPs. The rate of common mental disorder cases assessed by SRQ-20 was 65.1% among female IDPs compared with males (55.4%).

### 3.2. Traumatic Experiences

Most of the categories of experienced or witnessed traumatic events were similar between males and females (Table 2), and included murder, kidnapping, disappearance of others, forced to leave hometown, lack of housing or basic needs, forced isolation, and head injury and starvation. Males, in comparison with females, reported significantly higher rated of the two main categories of traumatic events: the first category included exposure to war-related physical and psychological threats, namely: (a) combat situation (explosion, artillery fire, shelling, or landmine), (b) being searched, (c) tortured (i.e., while in captivity one received deliberate and systematic infliction of physical and/or mental suffering), (d) oppressed because of ethnicity, religion, or sect, or (e) forced to change religion. The second category of traumatic events was witnessing destruction of personal properties, religious, or residential areas, namely (a) witnessing the desecration or destruction of religious shrines or places of religious instruction, (b) witnessing shelling, burning, or razing of residential areas or marshlands, or (c) witnessing rotting corpses. The mean score of exposure to a combat situation, torture, or oppression was significantly higher for males (M = 1.92, SD = 0.87) than for female IDPs (M = 1.69, SD = 0.89; t = 3.718, *p* < 0.001). Males also reported higher exposure to events concerning destruction of personal properties, religious, or residential areas (M = 0.54, SD = 0.85) than females (M = 0.29, SD = 0.62; t = 4.758, *p* < 0.001).

### 3.3. Prevalence of PTSD and Other Mental Health Symptoms

Table 3 presents the gender differences of Iraqi IDPs in the number of experienced or witnessed traumatic events and prevalence rates of PTSD symptoms and other mental symptoms. Male IDPs reported more traumatic events compared with female IDPs (t = 4.599, *p* < 0.001). The mean PTSD scores measured by 16 or 45 items of HTQ were not significantly different between both genders. There were no significant differences between the genders in terms of the PTSD symptom scores.

Symptoms of psychological distress as measured by GHQ-28 did not show significant differences between genders except somatic symptoms. The mean score of somatic symptoms was significantly higher in female Iraqi IDPs (M = 3.38, SD = 2.26) compared with male IDPs’ mean scores (M = 2.38, SD = 2.15; *p* < 0.001). Anxiety/insomnia, social dysfunction, and severe depression mean symptom scores were not significantly different between both gender groups.

The SRQ-20 mean total score was significantly higher among females (M = 9.52, SD = 4.54) compared with males (M = 8.45, SD = 4.39; *p* = 0.001). Higher mean scores of SRQ-20 depressive/anxious and somatic symptoms were found for female IDPs (M = 2.15, SD = 1.24 and M = 2.84, SD = 1.84, respectively) as compared with males (M = 1.62, SD = 1.08; *p* < 0.001 and M = 2.10, SD = 1.80, *p* < 0.001, respectively).

### 3.4. Predictors for PTSD

Binary logistic regression was carried out to study the possible predictors for probable PTSD among female and male Iraqi IDPs separately (Table 4). No significant association was found between any demographic characteristics including duration of stay in camps with the PTSD rate. The association between number of traumatic events and PTSD showed a marginal significant difference (*p* = 0.002) among females.

## 4. Discussion

In the present study, we examined whether experienced traumatic events, symptoms of PTSD and other mental health problems, and predictors for PTSD among Iraqi IDPs differed by gender. Although the number of experienced or witnessed traumatic events was higher among male than female Iraqi IDPs, the types of experienced traumatic events were similar with the exception that males reported more combat-related traumas and destruction of personal properties. Among respondents, 29.1% of female IDPs and 31.9% of male IDPs had a probable diagnosis of PTSD, which was not significantly different. Additionally, there was no statistically significant difference found for psychological distress reported by the two gender groups. PTSD scores were found to be similar across both genders. However, females suffered more from somatic and depressive/anxious symptoms. None of the sociodemographic factors, including the duration of camp stay, were associated significantly with PTSD diagnosis in both genders.

The number of experienced or witnessed traumatic events was found to be higher among male Iraqi IDPs compared with females. Our findings are in line with the scientific literature on the epidemiology of exposure to traumatic events worldwide which showed that females experienced fewer traumatic events of all kinds than males, except sexual violence [47,48]. It is also supported by findings from Iraq which showed that male Kurdish refugees experienced more traumatic events compared with females [36]. Further, the respondents reported similar common traumatic events with the exception that males reported higher exposure to combat situation, torture, oppressions, and destruction of personal properties. ISIS fighters used killing and violent murder as a common way of torture of males [5,48].

Contrary to previous findings showing a higher prevalence of PTSD in females than males [23,48,49,50,51], we did not find a significant difference between the genders for PTSD prevalence. This is remarkable, although some other studies in refugees and IDPs have failed to replicate the significant male–female difference for PTSD prevalence [13,52], including another independent study in Kurdish refugees from Syria resettling in Iraqi Kurdistan camps [36]. Some other studies on refugees or IDPs are in line with this finding. The comparable rates of PTSD for females and males in our and similar studies can be explained in various ways. Firstly, although it has been assumed that within the general population females are at a higher risk for PTSD than males [23], this may be different for IDPs who were exposed to numerous severe traumatic events in a short time period, such as during war or mass conflicts. In our study, most of the common traumatic events experienced by both genders were similar but the male IDPs reported higher exposure to war-related traumatic content or were an impact of the combat situation including torture, oppressions, destroying of personal properties, or destroying of religious and residential areas. This means that males in our study sample have experienced more severe types of traumatic events while females in our study hardly reported any sexual violence, which is mostly associated with high levels of PTSD [13]. It is unclear however, whether female IDPs may have underreported sexual violence due to shame or stigma in our study. Secondly, the different results from different studies can be explained by sociocultural differences within IDP and refugee populations in terms of their responses to traumatic events.

Our study showed also that females reported more somatic, depressive, and anxiety symptoms compared with male IDPs. This was in line with the results of a study on the asylum seekers and refugees from Chechnya, Afghanistan, and West Africa in which women reported more somatic symptoms and emotional outbursts than men [16]. Somatic symptoms were found to be associated with PTSD [53]. Among Asian populations, victims of traumatic events more commonly express distress as somatic symptoms than as other type of psychiatric symptoms [52], which may be related to culture and stigma [54]. Females usually seek social support more than males, and lacking of it during or after displacement times being the most coherent prognosticator of negative consequences of traumas [23]. The higher scores of depression thoughts and anxiety symptoms among females in our study was supported by many other studies [13,16,50,55].

This study was not free from limitations. First, we included participants from only one camp, whereas there are seventeen camps for IDPs in the Duhok province. Second, we relied on self-report, and we cannot preclude that specific events may have been underreported, such as sexual violence, which is usually associated with stigma. Despite these limitations, the strengths of this study are that we included a large sample of Iraqi IDPs assessed in a relatively short period after exposure to ISIS attacks and displacement (less than 1 year).

Further research is needed to focus more on how the IDP males and females react differently to traumatic events. Further, prospective or longitudinal studies are needed to examine which factors predict the persistence of PTSD and other common mental health problems over time. This may provide important information to develop culturally sensitive interventions to address mental health problems among both female and male Iraqi IDPs.

## 5. Conclusions

The findings of this study, carried out relatively shortly after exposure to ISIS attacks, found that males are more likely to experience traumatic events than females. Further, we found that probable rates of CMDs and PTSD among male and female IDPs in Iraq are substantial, with males and females equally affected. The results of our study show that effective mental health and psychosocial support strategies should be provided to Iraqi IDPs with PTSD and common mental disorders. Targeted and scalable culturally sensitive interventions should be implemented by local mental health authorities, human rights and especially non-governmental organizations to address these symptoms and improve overall wellbeing and functioning [56].

## Figures and Tables

**Table 1 ijerph-18-09779-t001:** Sociodemographic characteristics and PTSD and psychological distress prevalence rates according to gender. (*n* = 822).

Sociodemographic Characteristics	Females *n* (%)	Males *n* (%)	X^2^	df	*p*
**Total**	358 (43.6)	464 (56.4)			
**Age**			1.117	2	0.572
18–40 years old	271 (75.7)	347 (74.8)			
41–64 years old	68 (19)	98 (21.1)			
65 years old or more	19 (5.3)	19 (4.1)			
**Religion**			1.714 ^a^	2	0.461
Yazidi	355 (99.2)	459 (98.9)			
Muslim	1 (0.3)	4 (0.9)			
Christian	2 (0.6)	1 (0.2)			
**Marital Status**			0.556	1	0.456
Unmarried	85 (23.7)	100 (21.6)			
Married	273 (76.3)	364 (78.4)			
**Educational Level**			70.931	3	<0.001 *
Illiterate	218 (60.9)	146 (31.5)			
Educated	140 (39.1)	318 (68.5)			
**Employment**			72.415	1	<0.001 *
Unemployed	351 (98)	360 (77.6)			
Employed	7 (2)	104 (22.4)			
**Number of Siblings**			7.192	3	0.027
1–4 siblings	86 (24)	81 (17.5)			
5–8 siblings	141 (39.4)	219 (47.2)			
9+ siblings	131 (36.6)	164 (35.3)			
**Camp Stay in Months**			4.100	2	0.129
1–3 months	7 (2)	4 (0.9)			
4–6 months	17 (4.7)	13 (2.8)			
7–10 months	334 (93.3)	447 (96.3)			
**PTSD (by HTQ)**	104 (29.1)	148 (31.9)	0.770	1	0.212
**Psychological Distress (by GHQ-28)**	311 (86.9)	386 (83.2)	2.125	1	0.170
**Common Mental Disorders (by SRQ-20)**	233 (65.1)	257 (55.4)	7.891	1	0.005

PTSD, PTSD—first 16 questions of HTQ part IV mean >2.5; psychological distress, GHQ-28 > 4; common mental disorder, SRQ-20 > 7; X^2^, chi-squared test; ^a^ Fisher’s exact test; df, degree of freedom; *p*, *p*-value, * *p* < 0.002.

**Table 2 ijerph-18-09779-t002:** Gender differences in exposure to common traumatic events.

Traumatic Event	*n* of Events	Females		Males		t	df	*p*
		M	SD	M	SD			
Exposure to combat situation, torture, or oppressions	5	1.69	0.891	**1.92**	**0.867**	3.718	819	<0.001 *
Murder, kidnapping, disappearance of others	6	0.95	1.146	1.16	1.280	2.354	818	0.019
Forced to leave hometown	2	1.89	0.420	1.90	0.392	0.196	818	0.845
Lack of housing or basic needs	2	1.73	0.593	1.66	0.623	−1.486	819	0.138
Destruction of personal properties, religious, or residential areas	3	0.29	0.617	**0.54**	**0.854**	4.758	817	<0.001 *
Forced isolation	3	2.76	0.573	2.75	0.536	−0.252	820	0.801
Head injury and starvation	6	1.57	0.685	1.61	0.718	0.677	819	0.499

t, independent sample *t*-test, *p*, *p* value, * *p* < 0.002; bold face: statistically significant higher means than the opposite gender scores.

**Table 3 ijerph-18-09779-t003:** Number of traumatic events and prevalence rates of PTSD symptoms and other mental symptoms by gender.

Mental Symptoms	Range	Females		Males		t (*df* = 820)	*p*
		M	SD	M	SD		
**Total Number of Traumatic Events**	0–43	9.57	2.628	**10.53**	**3.192**	4.599	<0.001 *
**PTSD (HTQ-16)**	1–4	2.28	0.490	2.25	0.525	0700	0.484
**PTSD (HTQ-45)**	1–4	2.08	0.461	2.09	0.453	−0.152	0.880
(1) Intrusions	1–4	2.66	0.583	2.60	0.607	1.346	0.179
(2) Avoidance	1–4	3.21	0.823	3.12	0.892	1.428	0.154
(3) Numbing	1–4	1.70	0.652	1.83	0.644	−2.847	0.005
(4) Hyperarousal	1–4	2.18	0.680	2.05	0.709	2.638	0.008
**Total GHQ-28**	0–28	10.88	5.447	9.84	4.918	2.858	0.004
(1) Somatic symptoms	0–7	**3.38**	**2.259**	2.38	2.145	6.465	<0.001 *
(2) Anxiety/insomnia	0–7	3.22	1.942	2.83	2.008	2.827	0.005
(3) Social dysfunction	0–7	1.80	1.793	1.99	1.891	−1.480	0.139
(4) Severe depression	0–7	2.47	1.485	2.64	1.343	−1.648	0.100
**Total SRQ-20**	0–20	**9.52**	**4.538**	8.45	4.389	3.428	0.001 *
(1) Depressive/anxious	0–4	**2.15**	**1.237**	1.62	1.079	6.558	<0.001 *
(2) Somatic symptoms	0–6	**2.84**	**1.838**	2.10	1.798	5.807	<0.001 *
(3) Reduced vital energy	0–6	2.88	1.798	2.91	1.794	−251	0.802
(4) Depressive thoughts	0–4	1.65	1.115	1.81	1.094	−2.086	0.037

PTSD, post-traumatic stress disorder; HTQ, Harvard trauma questionnaire; GHQ, General health questionnaire; SRQ, Self-Reporting Questionnaire; t, independent sample *t*-test; *p*, *p* value, * *p* < 0.002; bold face: statistically significant higher means than the opposite gender scores.

**Table 4 ijerph-18-09779-t004:** Predictors of PTSD among male and female Iraqi IDPs.

Demographic Variables		PTSD Cases		Odd Ratios (OR)	
		Females *n* (%)	Males *n* (%)	Females (95%CI)	Males (95%CI)
Age					
	18–40 years	72 (69.2)	113 (76.4)	Ref	Ref
	41–64 years	24 (23.1)	29 (19.6)	1.45 (0.78–2.71)	0.70 (0.42–1.19)
	65+ years	8 (7.7)	6 (4.1)	2.28 (0.81–6.43)	0.77 (0.27–2.18)
Marital Status					
	Unmarried	30 (28.8)	21 (14.2)	Ref	Ref
	Married	74 (71.2)	127 (85.8)	0.61 (0.34–1.11)	2.15 (1.22–3.78)
Education					
	Illiterate	74 (71.2)	59 (39.9)	Ref	Ref
	Some education	30 (28.8)	89 (60.1)	0.46 (0.27–0.81)	0.62 (0.40–0.95)
Work Status					
	Unemployed	4 (3.8)	29 (19.6)	Ref	Ref
	Employed	100 (96.2)	119 (80.4)	0.25 (0.05–1.29)	1.30 (0.79–2.14)
*n* of Siblings					
	1–4 siblings	14 (13.5)	24 (16.2)	Ref	Ref
	5–8 siblings	47 (45.2)	69 (46.6)	2.54 (1.24–5.19)	1.19 (0.67–2.14)
	9+ siblings	43 (41.3)	55 (37.2)	2.41 (1.16–4.99)	1.28 (0.70–2.32)
Camp stay		-	-	1.50 (1.09–2.07)	1.27 (0.98–1.66)
*n* of traumas		-	-	1.16 (1.05–1.27)	1.01 (0.95–1.07)

*p*, *p*-value is statistically significant results when *p* < 0.002; OR, odds ratio; CI, confidence interval.

## Data Availability

The data that support the findings of this study are available on request from the corresponding author. The data are not publicly available due to their containing information that could compromise the privacy of research participants.

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
