# Peer review of "Gender Differences in Traumatic Experiences, PTSD, and Relevant Symptoms among the Iraqi Internally Displaced Persons"

_ijerph, 2021, doi:10.3390/ijerph18189779_

Round 1
Reviewer 1 Report
I read your manuscript with interest since you are addressing a significant issue - ie: gender differences in traumatic experiences, PTSD, and rele-2vant symptoms among the Iraqi internally displaced persons
The last part of sentence: “Explanations for the higher risk of PTSD in women than men are that the type of traumatic experiences may differ between males and females, with females more often exposed to high-impact interpersonal, sexual or gender-based trauma, and at a younger age than males” is difficult to understanding.
In the sentence “At first, the sample size was calculated according to this formula n = {Z2 p(1-p)}/e2 where (z) is the critical standard score, (p) is population proportion. (e) is margin of error” - the dot is useless, consider square brackets and superscript
Table 1. The Titles of tables and tables should be at the same page.
Table 2. Consider describing which traumatic events fell into the groups of traumatic events mentioned in Table 2, especially where the differences were statistically significant.
Higher mean scores of SRQ-20 depressive/ anxious and somatic symptoms were found for female IDPs as compared to males (M = 2.15, SD = 1.237; p < 0.001 and M = 2.84, SD = 1.838, p < 0.001 respectively): What is the score of depressive / anxious and somatic symptoms in the group of men?
Table 4. The entire table should be on one page.
Reviewer 2 Report
I have uploaded my comments on the paper which should be shared with authors.

Reviewer 3 Report
The article presented here is very interesting. However, considering the relevance of the topic and its topicality, it is necessary that the theoretical framework supporting the study is equally adequate.
For the time being, it is necessary to update the references, put more from 2020 and 2021. In addition, the DOI is missing in some of them.
As for the theoretical framework, it is superficial and needs to be improved. It is necessary to deepen and clarify what is the reality and situation of your group... What is the situation that these women live in the refugee camps? The examples you give of trauma and post-traumatic reactions are rather general.
This also impacts on the analysis of the results, which must be related to the theory... as the theoretical framework is so superficial, the analysis is left lacking in this relationship.
The conclusions are equally superficial. Conclusions should pay attention to the article's strengths and findings, as well as its limitations that may provide room for further research.
Round 2
Reviewer 3 Report
We appreciate the changes made.
The theoretical framework has been considerably strengthened, with much more up-to-date references that help to make the text more academic.
Similarly, the conclusions have been considerably improved.
I still believe that the theoretical framework is somewhat limited. It already has good references, but the section is still short and shallow. Still, I think it is sufficient.
The conclusions have improved and allow a better appreciation of the closure of the text.